# Objective Assessment of Upper-Extremity Motor Functions in Spinocerebellar Ataxia Using Wearable Sensors

**DOI:** 10.3390/s22207993

**Published:** 2022-10-20

**Authors:** Reza Mohammadi-Ghazi, Hung Nguyen, Ram Kinker Mishra, Ana Enriquez, Bijan Najafi, Christopher D. Stephen, Anoopum S. Gupta, Jeremy D. Schmahmann, Ashkan Vaziri

**Affiliations:** 1BioSensics LLC, 57 Chapel St, Newton, MA 02458, USA; 2Interdisciplinary Consortium on Advanced Motion Performance (iCAMP), Michael E. DeBakey Department of Surgery, Baylor College of Medicine, Houston, TX 77030, USA; 3Ataxia Center, Department of Neurology, Massachusetts General Hospital, Harvard Medical School, 100 Cambridge St, Boston, MA 02115, USA

**Keywords:** movement disorder, telemedicine, care in place, remote patient monitoring, digital biomarker, scale for the assessment and rating of ataxia, dysdiadochokinesia

## Abstract

The study presents a novel approach to objectively assessing the upper-extremity motor symptoms in spinocerebellar ataxia (SCA) using data collected via a wearable sensor worn on the patient’s wrist during upper-extremity tasks associated with the Assessment and Rating of Ataxia (SARA). First, we developed an algorithm for detecting/extracting the cycles of the finger-to-nose test (FNT). We extracted multiple features from the detected cycles and identified features and parameters correlated with the SARA scores. Additionally, we developed models to predict the severity of symptoms based on the FNT. The proposed technique was validated on a dataset comprising the seventeen (*n* = 17) participants’ assessments. The cycle detection technique showed an accuracy of 97.6% in a Bland–Altman analysis and a 94% accuracy (F1-score of 0.93) in predicting the severity of the FNT. Furthermore, the dependency of the upper-extremity tests was investigated through statistical analysis, and the results confirm dependency and potential redundancies in the upper-extremity SARA assessments. Our findings pave the way to enhance the utility of objective measures of SCA assessments. The proposed wearable-based platform has the potential to eliminate subjectivity and inter-rater variabilities in assessing ataxia.

## 1. Introduction

Spinocerebellar ataxia (SCA) is characterized by motor dysfunction caused by degenerative changes in the cerebellum [1]. SCA-related motor dysfunction often manifests as abnormal limb coordination, speech difficulties, oculomotor abnormalities, and gait disorder, with impaired postural control during walking or standing [1,2,3,4]. Over time, several motor tasks were designed and standardized to evaluate these motor disturbances in terms of accuracy, timing, rhythmicity, and stability, and corresponding impairments were termed dysmetria, dyssynergia, and dysrhythmia [5]. Among current methods to assess SCA, the Scale for the Assessment and Rating of Ataxia (SARA) is widely used and includes the finger chasing test (FCT), finger-to-nose test (FNT), and alternating hand movement assessment for dysdiadochokinesia (DDKT) [2,6]. In addition to the SARA, other assessment scales, including the Brief Ataxia Rating Scale (BARS) and the International Cooperative Ataxia Rating Scale, can effectively determine specific aspects of disease severity. However, these scales must be administered by a neurologist with particular expertise in ataxia and are subjective [5,7,8]. For instance, the SARA stipulates the overshoot or undershoot distance between the patient’s finger and the clinician’s finger in the FCT. This evaluation and scoring system is prone to bias and inaccurate assessment owing to its subjective evaluation, and it is potentially compounded if the same clinician does not consistently evaluate the patient. The objective measurements can provide more consistent evaluations and are crucial when specialists are unavailable.

Moreover, objective measures may reveal patterns of motor control dysfunction below the threshold of detection even by subspecialty-trained clinical ataxiologists. More specifically, quantitative assessments in pre-symptomatic or early disease stages in neurodegenerative diseases may detect subtle changes in motor abilities that would not be sufficient to change the ordinal rating scale score from one severity level to the next on available clinical rating scales [9,10,11]. Therefore, such an approach may facilitate repeated assessments that can more accurately and reliably detect changes over time. This can lead to improved outcome measures and compensates for the impact of biological variability (i.e., random fluctuations in motor response within the subject) in single time-point assessments. Several studies have proposed the objective assessment of ataxia [12,13]. For instance, a push-button system to evaluate the variation in the timing of ataxic movements was considered for the finger tapping test [14], or using optoelectronic devices (e.g., VICON, Kinect, or video camera) to assess delay in initiating movement and accuracy in reaching the target. Recently, inertial measurement units (IMUs) were used to quantify the FNT [3,4] and DDKT [15] performance. The working principle of these IMUs is based on estimating movement using onboard sensors, including an accelerometer, magnetometer, or gyroscope. The benefit of IMUs over other sensing modalities is the precise measurement of angular acceleration and velocity with minimal preparation and no interference with the motor examination.

Although these systems can provide the intended information for specific disease severity aspects, they have their limitations. For instance, using a system consisting of special video cameras, touch screens, and a full-size computer to sync various components to track a patient’s finger trajectory in the FCT may provide precise quantification of movement dynamics. However, it still requires the patients to travel to a clinic because such a data acquisition system is neither widely accessible nor easy to operate for non-specialists. The other challenge, especially in remote assessments, is to devise an automated technique in all its subcomponents. For instance, the average cycle time for a cyclic test such as the FNT, which is the focus of this study, is a widely used feature to analyze the results of this test. An objective assessment algorithm should detect cycles to extract relevant features from the associated data within each cycle and predict the severity score. Such a comprehensive procedure represents an important gap in the literature.

The objectives of this study were to develop a wearable-based solution that can objectively assess upper-extremity motor symptoms using wearable sensors. In addition, we investigated possible redundancies in the upper-extremity SARA assessments. In doing so, we made the following contributions: First, we developed an algorithm for detecting/extracting the cycles of the FNT. We also extracted multiple features from the detected cycles, investigated their correlation with the SARA scores, and developed models to predict the severity of symptoms based on the FNT. Second, we studied the dependency between the upper-extremity SARA tests through correlation analyses and building predictive models to estimate the severity of symptoms based on the FCT and DDKT using the extracted features from the FNT. Our findings pave the way for the enhancement of the utility of objective measures of SCA assessments using wearable sensors and complement our previous study focused on objective assessments of gait and balance in SCA [11].

## 2. Materials and Methods

### 2.1. Participants

This study protocol was approved by the Partners HealthCare System Institutional Review Board. All participants signed the informed consent form before participating in the study. Seventeen participants were recruited from the Massachusetts General Hospital (MGH) Ataxia Center and studied at the MGH Neurological Clinical Research Institute (NCRI). Fourteen patients with a genetically confirmed diagnosis of SCA (SCA1: *n* = 1, SCA2: *n* = 3, SCA3: *n* = 5, SCA6: *n* = 5) and three healthy controls were enrolled. A subspecialty ataxiologist (JDS) determined the eligibility for the study participation. Exclusion criteria were less than 18 or more than 75 years of age, inability to walk independently even with a walker (SARA or BARS gait score > 6), inability to comply with all study activities, or unwillingness to provide informed consent. The inability to walk was an exclusion criterion because this study was part of a comprehensive investigation of both upper and lower limb dysfunction and gait.

### 2.2. Clinical Assessments

During the clinical visit, participants underwent a standardized ataxia evaluation using the SARA and BARS version 2 (BARS2, St. Petersburg, Russia) which utilizes half-points according to designated descriptors included in the scale [7,9]. All participants were equipped with 2 IMUs (LEGSys^TM^, BioSensics, Newton, MA, USA) on their wrists during the assessments (Figure 1). Each sensor consists of a triaxial accelerometer and a triaxial gyroscope with a sampling frequency of 100 Hz. The dynamic ranges of the accelerometer and gyroscope are, respectively, ±2 g and ±2000 deg/s, where g is the gravitational acceleration. The sensors were placed with the help of the research staff, and two clinical assessments of SARA and BARS2 were performed, with a duration of 30 min in-between. The sensors were secured over the wrist using elastic Velcro to avoid any shift in the sensor placement. The assessments were also video recorded. The SARA and BARS2 scores were conducted and rated in person by a neurologist (J.D.S). The video recordings were later reviewed and scored according to the BARS2 and SARA by two additional ataxia specialists (C.D.S, A.S.G) to examine inter-rater reliability. Each rater was blinded to the scores given by other raters.

This study only assessed the upper-extremity SARA assessment of the FNT, FCT, and DDKT. The FNT measures how smooth and coordinated the upper-extremity movement is by having the subject perform a cyclic task. In this task, the clinician holds his/her finger at 90% of their arm’s length from the patient, and the patient is instructed to touch the clinician’s finger and then his/her nose in several consecutive trials. For the FCT, the clinician performs five consecutive sudden and fast pointing movements in unpredictable directions in a frontal plane. The amplitude and frequency of the movements are, respectively, about 30 cm and 1 movement per 2 s. The patient is asked to follow the movements with his/her index finger as fast and precisely as possible, and the average performance of the last three iterations is rated. Finally, the DDKT measures how quickly and accurately one can repeat a rapid alternating pronation/supination task by asking the patient to tap the palm of one hand to his/her thigh, raise and flip the hand, tap the back of the hand to the thigh, raise and flip the hand, and repeat this cycle ten times with the regularity of the movement and time taken to perform 10 iterations recorded.

### 2.3. FNT Cycle Detection Algorithm

In this study, we focused on the cyclic FNT task. The objective of the proposed algorithm was to extract each cycle of the FNT from the sensor-derived kinematic signal. The proposed algorithms detect each cycle and identify different phases within each cycle as follows:Phase 1 (Decline): from the time instant that the patient’s finger starts moving from the clinician’s finger until reaching their nose.Phase 2 (Pause at the nose): the period between the instant that the patient’s finger reaches their nose until they start moving towards the clinician’s finger.Phase 3 (Rise): from the instant that the patient’s finger starts moving from their nose until they reach the clinician’s finger.Phase 4 (Pause at finger): the period between the instant that the patient’s finger reaches the clinician’s finger until they start moving towards their nose.

The duration of phases 2 and 3 may be zero if the participant does not pause when they reach their nose or the clinician’s finger. The segmented signal is then used to extract features dependent on the disease severity.

We used the Euler angles to extract the four phases of each cycle derived from the quaternion values provided by the sensors. First, the sensor quaternion is calibrated for the local fixed frame of the clinician’s finger, and then the Euler angles are estimated using the calibrated quaternion. A calibrated time series for the right hand of a healthy individual and a participant with SCA is shown in Figure 2a. Given that the quaternions were calibrated at the clinician’s finger and assuming that the finger does not move during the test, the angle takes its maximum and minimum values when the participant reaches his nose and the clinician’s finger, respectively. These local extrema are easy to capture for the healthy participant because each cycle contains only one maximum and one minimum. However, multiple local extrema may exist in the angle time history of each FNT cycle for the participant with SCA.

To further clarify this, consider the angle signal of one FNT cycle for an SCA participant in Figure 2a, which is magnified in Figure 2b. Instead of a single maximum or minimum, we observe peaks and valleys with multiple local extrema. The valley starts at point A where the participant is at the clinician’s finger. There is no significant change in angle from point A to B, where the valley ends. This part is associated with the pause at the clinician’s finger, i.e., phase 4, and the local extrema between A and B are due to tremor. A rapid change in angle starts at B and ends at C, corresponding to the movement between the clinician’s finger and the participant’s nose, i.e., phase 1. The CD segment is similar to AB but occurs at the participant’s nose, i.e., phase 2.

Finally, there is a rapid change in angle from D, which corresponds to phase 3, until the participant reaches the clinician’s hand again at E. Therefore, by detecting the A, B, C, D, and E points in each cycle of the signal, the four phases of the FNT motion can be detected. Note that the direction of rotation from finger to nose is counterclockwise for the right hand, while it is clockwise for the left hand; thus, the left-hand measurements can be similarly analyzed after changing the sign of the Euler angles.

### 2.4. Signal Processing

#### 2.4.1. Data Pre-Processing

The measured signals were filtered with a sixth-order band-pass Butterworth filter with cut-off frequencies of 0.2 and 20 Hz. The lower bound cut-off was used to minimize the drift effects, and the exclusion of high-frequency components was due to the restriction in the maximum frequency that can be generated by human movements [16].

#### 2.4.2. Feature Extraction

Two types of features are extracted from the sensor data. The first type is extracted from each FNT cycle and was extracted using the cycle detection algorithm, and the second type of features is extracted from the entire signal of a complete FNT task. We only considered the clinically relevant features and categorized them into five phenotypes: timing, speed, variability, rhythmicity, and steadiness. All features considered in this study are summarized in Table 1. In this table, |.| is the absolute value; motion intensity (MI) of the angular velocity (AV) and linear acceleration (LA) are
$${\mathrm{MI}}_{\mathrm{AV}}=\sqrt{\dot{\theta_{x}^{2}}+\dot{\theta_{y}^{2}}+\dot{\theta_{z}^{2}}}$$
(1)$${\mathrm{MI}}_{\mathrm{LA}}=\sqrt{{\ddot{x}}^{2}+{\ddot{y}}^{2}+{\ddot{z}}^{2}}$$
where *x*, *y*, and *z* are displacement components along the sensor’s local axes, the double dot is the second time derivative, (*θ*) is the angular velocity, its subscripts represent the axis of rotation, and a single dot is the first time derivative. The speed metric is defined as the mean value of speed divided by the maximum speed [17], where speed can be any member of $\left\{\dot{\theta_{x}},\;\dot{\theta_{y}},\;\dot{\theta_{z}},\;{\mathrm{MI}}_{\mathrm{AV}}\right\}$. The jerk index is the root-mean-square value of the derivative of acceleration data, i.e., the jerk, normalized by the maximum value of the velocity data [1,15].

Extracting these features from the angular velocity and linear acceleration in all directions and the motion intensities results in 140 features. Note that there are two types of features in general; the first type has a single value for an entire FNT, such as the maximum and range of AV, and the second type has a single value for each cycle and, hence, multiple values for each test, such as the cycle time. Herein, we call the second type of features cycle-based features. To combine the two types of features, first, an interquartile range (IQR) outlier detection was used to remove the outliers from the cycle-based features. The IQR, denoted by $\delta_{q}$, is the range between the first and the third quartiles, namely, $Q_{1}$ and $Q_{3}$, i.e., $\delta_{q}=Q_{3}-Q_{1}$. The data points which fall outside of $\left[Q_{1}-1.5\;\delta_{q},Q_{3}+1.5\;\delta_{q}\;\right]$ interval were considered outliers and removed. The average of the remaining values, along with the features extracted from the entire signal, are considered the feature vector for each test.

## 3. Statistical Analysis and Results

### 3.1. Reliability Assessment

The intraclass correlation coefficient (ICC) was performed to determine the test-retest and inter-rater reliability of the FNT Score from SARA and BARS scales (ICC; two-way random, absolute agreement). The ICC reflected the degree of correspondence and the degree of agreement between the scorings by the three raters. Reliability was considered good if the ICC was greater than 0.75 and fair if the ICC was between 0.40–0.75 [18]. Table 2 shows excellent intra-rater reliability as the ICC was in the range of 0.94–0.99. Similarly, the ICC was between 0.93 and 0.99, indicating excellent inter-rater reliability.

### 3.2. Cycle Detection Algorithm Evaluation

The proposed cycle detection algorithm was tested on the collected dataset, and the results were compared with a ground truth obtained by extracting each cycle’s start/end from the recorded videos. Figure 3a shows the scatter plot of the extracted cycle duration using the algorithm vs. the gold standard for 535 cycles. The correlation between the estimated and gold standard cycle duration is 97.6%. The associated Bland–Altman plot is also shown in Figure 3b, and implies a high agreement between the algorithm results and the ground truth.

### 3.3. Sensitivity Analysis of the Extracted Features and Their Correlation with SARA Scores

A Spearman rank correlation analysis was conducted to identify the features with the highest correlation to the FNT SARA score. A Mann–Whitney signed-rank test was used to determine if the extracted features are statistically significant for discriminating between the control and SCA participants. The features that were correlated with the FNT SARA score and statistically significant (<0.05) according to the Mann–Whitney test are listed in Table 3. The Cohen’s *d* effect size, the mean value, STD, and the mean difference percentage of the selected features for the two control and SCA participants are also shown in Table 3. It follows from the results that specific features, such as the pause duration or the STD of the rise time, are strongly correlated with the SARA score and significantly different between the control and individuals with SCA. The duration of phases 2 and 4 of each cycle, i.e., pause at nose and finger, respectively, are almost negligible for control participants, while these phases, on average, take more than 17% of the test duration in subjects with SCA.

### 3.4. Dimensionality Reduction and Classification

The available dataset is relatively small and highly unbalanced. Having 17 participants who are tested two times for each hand and noting that sometimes the rater asks a participant to redo the test in either of the trials, the dataset consists of 34 and 31 distinct tests for the first and second trials, respectively. The distribution of SARA scores in this dataset is summarized in Table 4a. 

For training and predicting the SARA scores, the data points associated with SARA scores of two and three were grouped and categorized as moderate SCA to make the dataset more balanced. SARA scores of zero and one were considered control and mild SCA categories, respectively. The distribution of the dataset in these categories is illustrated in Table 4b. In this study, we trained classifiers to predict the categories of Table 4b, which are closely related to the SARA score.

We used a gradient boosting classifier (GBC), logistic regression (LR) with elastic-net regularization, and support vector classifier (SVC) with radial basis function (RBF) kernel to predict the severity of SCA in our dataset. The models were trained using the data from trial 1, and their performances were evaluated using the data from trial 2. Principal component analysis (PCA) was conducted on the feature matrix to reduce the dimensionality and manage collinearity between features. The GBC was trained on both the original features and all principal components (PCs). The LR was trained using the PCs only due to the sensitivity of this model to features’ collinearity. Finally, the SVC was trained along with a random subspace feature selection scheme using the PCs. All the hyper-parameters of these models were optimized through cross-validation on the data of trial 1. Table 5 compares the performance of these models when evaluated on the unseen dataset of trial 2. The performances of the GBC using the PCs and the original features were similar; therefore, the result of only one of the GBC models is reported in this table. The weights used for the “weighted average” column are proportional to the number of sample points for each class to account for the class imbalance data. It follows from the prediction results that the severity of the SCA can be accurately predicted from the extracted features, even with a small and unbalanced dataset.

### 3.5. Redundancies of Upper-Extremity SARA Evaluation

Although each motor task (i.e., FNT, FCT, or DDKT) provides unique information about movement dysfunction, there might be an inherent redundancy among these motor tasks [19,20]. The purpose of estimating the correlations was to capture the inherent redundancies among the tests (i.e., FNT. FCT, and DDKT) through correlation analyses. However, the SARA scores of the FCT and DDKT were used instead of the FNT. Table 6 lists the features that are correlated with the upper-extremity SARA scores based on the Spearman correlation analysis with a significance level of 0.05. The correlation coefficients and associated *p*-values are presented, except for the features with *p*-values > 0.05, i.e., not significantly correlated with a specific test. For instance, “STD of cycle time” is correlated with the FNT and DDKT but not with the FCT.

To better capture the dependencies of the upper-extremity SARA tests, classification analyses were performed to predict the SCA severity of the FCT and DDKT using the features that were extracted from the FNT. The distribution of the sample points for the FCT and DDKT is presented in Table 7. The classification results are illustrated in Table 8 and Table 9. It follows from the results that the severity of the DDKT and FCT can be predicted with 71% and 81% accuracy via an SVC model trained on the features that were extracted from the FNT.

The results show a strong dependency between the FNT features and the SARA scores of DDKT and the FCT, although they may not be significantly correlated. Additionally, this dependency can be captured by a nonlinear kernel such as the RBF. This demonstrates the redundancies in upper-extremity SARA tests for predicting the SARA scores or SCA severity despite their unique aspects in showing different types of disabilities. Note that we only used clinically meaningful features and a small dataset in this study; thus, the predictions are expected to be improved by using more advanced features on a larger dataset.

## 4. Discussion

We have developed a wearable-based solution for the objective assessment of the upper-extremity SARA test. Seventeen participants were recruited from the MGH Ataxia Center and studied in the MGH NCRI. Participants underwent a standardized ataxia evaluation using the SARA and BARS2 while wearing a single sensor on each wrist. We observed excellent inter-rater reliability for the SARA and BARS2 for all participants among the three raters, one of whom rated the participants in person and the others rated based on videos (Table 2). These results show consistency in the target class ratings.

A cycle detection algorithm was developed for extracting the cycles of the FNT from the sensor measurements and segmenting each cycle data into four main phases, i.e., decline, pause at the nose, rise, and pause at the finger. The algorithm extracted 140 features from the segmented and full sensor measurements. The estimated features were correlated with the SARA scores and could accurately discriminate the control and SCA severity groups related to the FNT. The benefit of the proposed objective assessment over conventional subjective evaluation includes the precise measurement of movement. Due to high sensitivity in tracking hand movement and detecting smaller vibrations (i.e., tremors), the proposed solution may identify subtle motor symptoms, which are often challenging to detect visually.

Unlike previous studies that used wearable-based validated gait assessment tools to assess SCA severity [11,21], the present study focused on upper-extremity motor examination by breaking down arm movement into discrete subcomponents using the wearable. This is a novel and valuable approach for representing and quantifying the ataxia phenotype, as gait assessment can only be administered for ambulatory individuals and requires dedicated facilities (e.g., an uninterrupted walkway or trained staff to ensure safety). The present study would contribute to validating the wearable-based upper-extremity motor assessment to assess SCA severity.

Furthermore, the dependency of the upper-extremity SARA tests was investigated via training machine learning algorithms to predict the SARA scores related to the FCT and DDKT using the features obtained from the FNT. The high accuracy of the SVC with the nonlinear RBF kernel implies that the upper-extremity SARA tests are dependent, although their linear correlation may be insignificant. The results suggest that the number of motor tasks could be potentially reduced while estimating the upper-extremity motor symptoms with acceptable accuracy. A reduced number of motor tasks may improve the feasibility of using the platform in a non-clinical setting and for remote patient monitoring. Digital health technologies are poised to become an integral part of modern health care, expedited by the surge in remote care necessitated by the COVID-19 pandemic [22,23,24,25,26,27,28,29,30,31]. The proposed solution paves the way to integrating digital health technology (i.e., wearables) to enhance the utility of objective measures of ataxia.

Previously, Kasyap et al. (2020) proposed a comprehensive approach to the evaluation of cerebellar ataxia by objectively assessing five domains (speech, upper limb, lower limb, gait, and balance) through the instrumented versions of nine bedside neurological tests using wearable IMUs and microphones [3]. Similarly, Tran et al. (2020) also proposed a comprehensive scheme for objective upper-body assessments of subjects with cerebellar ataxia [5]. Tran et al. used an IMU and a Kinect camera system to evaluate the FNT, DDKT, and FCT performance. While our results were comparable to these studies, we also addressed the redundancy of the upper-extremity assessment tasks and primarily focused on parameters estimated from the FNT. In addition, Oubre et al. extracted the movement elements from the participants’ accelerometry data during the FNT and employed machine learning algorithms. The classification models distinguished between ataxia and healthy controls, and ataxia and Parkinsonism phenotypes with areas under the receiver–operating curve of 0.96 and 0.86, respectively [13]. These findings support the evolving recognition that wearable-based systems may be helpful in managing movement disorders in the clinic or in home settings and potentially reduce observer bias in clinical trials. Consequently, the frequency of assessments may increase, which may compensate for the biological variability at single time points (same day, different times of day, weeks/months apart) in clinical care and clinical trial design.

A major limitation of the present study is the small sample size. While the high-performance accuracy of the current models is encouraging, the results need to be confirmed in a larger dataset. Furthermore, the models were trained on trial 1 and tested on trial 2. Since we have the same participants in trial 1 and trial 2, the model might have learned individual characteristics in addition to disease characteristics. A further limitation was the absence of late-stage severe SCA patients in whom somatosensory impairment, vestibular involvement, or other central nervous systems (CNS) lesions may contribute to the overall disability [5]. It is important to establish whether the factors differentiating SCA associated with other neural lesions might differ from “pure” cerebellar ataxia to produce a more precise means of assessing ataxia. This would be a subject of future studies, which can also compare the platform’s accuracy in differentiating ataxia from other neurodegenerative diseases (e.g., Parkinson’s disease). Similarly, more severe ataxia reflected by SARA FNT scores > 3 would be important in future studies. The benefit of including individuals with a low to moderate level of SCA clinical severity is to highlight the finding that objective assessment using the proposed platform can perhaps detect SCA in the early stages.

## 5. Conclusions

Our findings pave the way to enhancing the utility of objective measures of SCA assessments using wearable sensors, and complement our previous study focused on objective assessments of gait and balance in SCA. The proposed wearable-based platform has the potential to eliminate subjectivity and inter-rater variabilities in assessing ataxia. It can also facilitate frequent remote patient assessments that may compensate for the impact of biological variability in intermittent single visits. The findings of this study need to be validated in a larger sample.

## Figures and Tables

**Figure 1 sensors-22-07993-f001:**
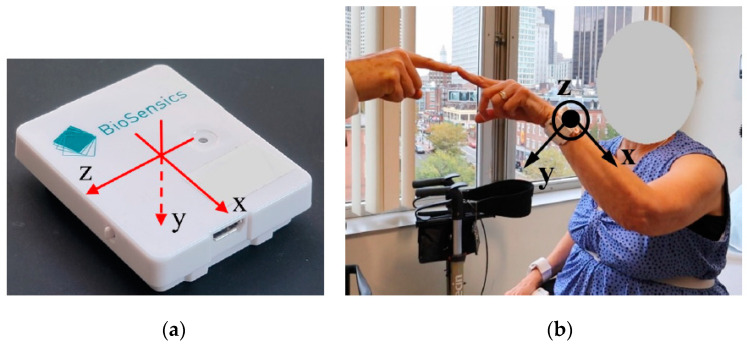
LEGSys^TM^ sensor used in this study. (**a**) Local axes of a LEGSys sensor, (**b**) wrist sensor with its axes shown during an FNT.

**Figure 2 sensors-22-07993-f002:**
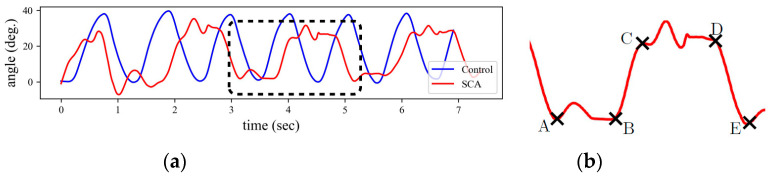
Euler angle in y-axis for right hand. (**a**) Comparison between the Euler angle time-history for a healthy individual and a participant with SCA. (**b**) Key points for identifying the four phases of each FNT cycle: point A is the time instant that the participant reaches clinician’s finger, point B corresponds to the time instant that the participant starts moving their hand towards the nose, point C is when they reach the nose, point D is the instant that they start moving towards the clinician’s finger, and E is the instant that they reach the clinician’s finger again.

**Figure 3 sensors-22-07993-f003:**
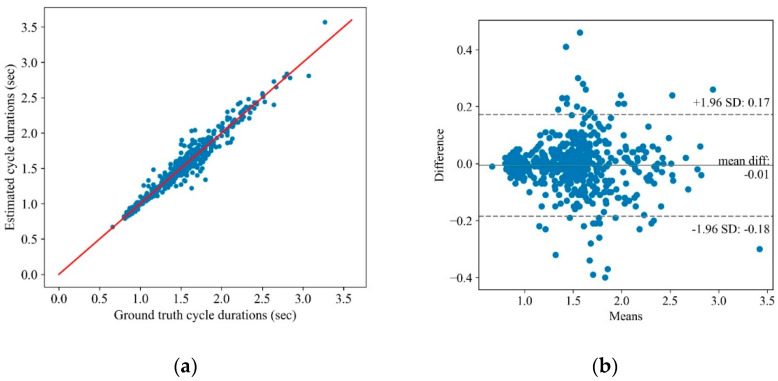
The proposed cycle detection algorithm results: (**a**) the scatter plot of the ground truth cycle duration on the x-axis vs. the estimated cycle duration via the algorithm on the y-axis and (**b**) the Bland–Altman plot of the mean vs. the difference in the ground truth cycle duration and the estimated one using the algorithm.

**Table 1 sensors-22-07993-t001:** List of all extracted features in this study.

Type	Phenotype	Features
Cycle-based	Timing	Cycle time (duration of a full cycle), decline/rise time (duration of phases 1 and 3, respectively), pause duration at patient’s nose (duration of phase 2), pause duration at rater’s finger (duration of phase 4), number of cycles per second.
	Variability	Standard deviation (STD) and coefficient of variation (CoV) of all timing features, STD of angular velocity (AV), and linear acceleration (LA)
	Speed	Maximum, minimum, range, mean value, and root-mean-square of motion intensity of AV and LA and all three components of the AV, |AV|, LA, |LA|, speed metric of AV and MI_AV_.
	Steadiness	Jerk index of the motion intensity of AV and LA and all three components of the AV and LA.
Entire signal	Rhythmicity	The first resonant frequency (RF) and the magnitude of resonant frequency (MR) of each component of AV and LA.
		The second RF and MR of each component of AV and LA.

**Table 2 sensors-22-07993-t002:** Inter- and intra-rater reliability of the FNT score from SARA and BARS.

Scores	Intra-Rater Reliability		Inter-Rater Reliability	
	SARA	BARS2	SARA	BARS2
FNT	0.94	0.97	0.94	0.93
Overall score	0.99	0.99	0.95	0.99

**Table 3 sensors-22-07993-t003:** Features with high correlation with the FNT SARA score and significant sensitivity to discriminate the control and SCA subjects.

		Mann–Whitney Test		Control/SCA Comparison		
Phenotype	Feature	*p*-Value	Cohen’s *d*	Control	SCA	Diff (%)
Timing	Cycle time (s)	<0.01	1.33	1.02 ± 0.18	1.61 ± 0.47	−58
	Cycle per second	<0.01	2.65	0.99 ± 0.15	0.66 ± 0.12	34
	Pause at finger (s)	<0.01	0.86	0.01 ± 0.03	0.15 ± 0.18	−1187
	Pause at nose (s)	<0.01	0.67	0.003 ± 0.008	0.12 ± 0.20	−4866
Variability	STD of cycle time (s)	<0.01	0.50	0.05 ± 0.02	0.18 ± 0.27	−224
	STD of rise time (s)	<0.01	0.95	0.04 ± 0.03	0.09 ± 0.06	−139
	STD of ${\dot{\theta }}_{y}$ (deg/s)	<0.01	1.88	111.5 ± 8.9	77.0 ± 19.5	31
	STD of $\left|{\dot{\theta }}_{y}\right|$ (deg/s)	<0.01	1.25	63.4 ± 4.0	47.9 ± 13.4	24
	STD of pause at finger (s)	<0.01	0.95	0.015 ± 0.034	0.096 ± 0.092	−547
	STD of pause at nose (s)	<0.01	0.42	0.006 ± 0.019	0.074 ± 0.176	−1231
	STD of total pause (s)	<0.01	0.59	0.025 ± 0.053	0.155 ± 0.238	−527
Speed	max $\left({\dot{\theta }}_{y}\;\right)$	<0.01	0.98	197.8 ± 20.1	155.3 ± 46.3	22
	Range of ${\dot{\theta }}_{y}$	<0.01	1.15	397.8 ± 33.0	309.9 ± 82.1	22
	Average of $\left|{\dot{\theta }}_{y}\right|$	<0.01	2.19	91.9 ± 9.8	60.2 ± 15.2	35
	RMS of ${\dot{\theta }}_{y}$	<0.01	1.90	112.6 ± 9.2	77.4 ± 19.7	31
Rhythmicity	First RF of ${\dot{\theta }}_{x}$	<0.01	1.67	0.98 ± 0.19	0.67 ± 0.17	31
	First RF of ${\dot{\theta }}_{y}$	<0.01	1.87	1.01 ± 0.19	0.67 ± 0.18	34
	First MR of ${\dot{\theta }}_{y}$	<0.01	2.47	1.6 × 10^−4^ ± 4 × 10^−3^	71 × 10^−4^ ± 4 × 10^−3^	57
	First RF of $\ddot{x}$	<0.01	0.73	1.17 ± 0.52	0.78 ± 0.52	33
Steadiness	Jerk index of ${\dot{\theta }}_{x}$	<0.01	1.79	27.0 ± 6.7	18.1 ± 4.4	33
	Jerk index of MI_AV_	<0.01	1.35	12.6 ± 3.6	8.8 ± 2.6	30
	Second RF of ${\dot{\theta }}_{x}$	<0.01	1.79	2.92 ± 0.57	1.57 ± 0.77	46

Values are presented as mean ± standard deviation.

**Table 4 sensors-22-07993-t004:** Distribution of SARA scores and severity class in the dataset.

(a) Data Points per SARA Scores						(b) Data Doints per SCA Severity Class			
	No. of Data Points per Score					No. of Data Points per Severity Class			
Trial	0	1	2	3	Total	Control	Mild SCA	Moderate SCA	Total
Trial 1	8	22	3	1	34	8	22	4	34
Trial 2	7	20	2	2	31	7	20	4	31

**Table 5 sensors-22-07993-t005:** Comparing different classifiers for predicting the severity of FNT.

Classifier	Metric	Control	Mild SCA	Moderate SCA	Accuracy	Weighted Average
SVC	Recall	0.86	1.0	0.75	0.94	0.94
	Precision	1.0	0.91	1.0		0.94
	Specificity	1.0	0.82	1.0		0.88
	F1-score	0.92	0.95	0.86		0.93
LR	Recall	0.71	1.0	0.25	0.84	0.84
	Precision	1.0	0.8	1.0		0.87
	Specificity	1.0	0.55	1.0		0.71
	F1-score	0.83	0.89	0.4		0.81
GBC	Recall	0.71	0.75	0.75	0.74	0.74
	Precision	0.55	0.83	0.75		0.76
	Specificity	0.83	0.73	0.96		0.78
	F1-score	0.62	0.79	0.75		0.75

**Table 6 sensors-22-07993-t006:** Correlation of the FNT features with the SARA scores of FNT, DDKT, and FCT.

		FNT		DDK		FCT	
Phenotype	Feature	ρ	*p*-Value	ρ	*p*-Value	ρ	*p*-Value
Timing	Cycle time (s)	0.70	<0.01	0.66	<0.01	0.46	<0.01
	Cycle per second	−0.70	<0.01	−0.66	<0.01	−0.46	<0.01
	Pause at finger (s)	0.57	<0.01	0.52	<0.01	0.39	<0.01
	Pause at nose (s)	0.59	<0.01	0.49	<0.01	-	-
	Total pause (s)	0.64	<0.01	0.57	<0.01	0.47	<0.01
Variability	STD of cycle time (s)	0.38	0.03	0.38	0.03	-	-
	STD of rise time (s)	0.39	0.02	0.43	0.01	-	-
	STD of pause at finger (s)	0.52	<0.01	0.44	0.01	0.44	0.01
	STD of pause at nose (s)	0.37	0.038	0.40	0.02	-	-
	STD of total pause (s)	0.47	<0.01	0.52	<0.01	-	-
	$\mathrm{STD}\;\mathrm{of}\;{\dot{\theta }}_{y}$ (deg/s)	−0.52	<0.01	−0.65	<0.01	-	-
	$\mathrm{STD}\;\mathrm{of}\;\left|{\dot{\theta }}_{y}\right|$ (deg/s)	−0.41	0.02	−0.55	<0.01	-	-
Speed	$\max\left({\dot{\theta }}_{y}\;\right)$	−0.41	0.02	−0.52	<0.01	-	-
	$\mathrm{Range}\;\mathrm{of}\;{\dot{\theta }}_{y}$	−0.38	0.03	−0.49	<0.01	-	-
	$\mathrm{Average}\;\mathrm{of}\;\left|{\dot{\theta }}_{y}\right|$	−0.57	<0.01	−0.66	<0.01	-	-
	$\mathrm{RMS}\;\mathrm{of}\;{\dot{\theta }}_{y}$	−0.53	<0.01	−0.64	<0.01	-	-
	$\mathrm{Max}\left(\left|{\dot{\theta }}_{y}\right|\right)$	-	-	−0.44	0.01	-	-
	$\mathrm{Max}\left(\ddot{x}\right)$	0.35	0.047	-	-	-	-
Rhythmicity	$\mathrm{First}\;\mathrm{RF}\;\mathrm{of}\;{\dot{\theta }}_{x}$	−0.64	<0.01	-0.65	<0.01	−0.40	<0.01
	$\mathrm{First}\;\mathrm{RF}\;\mathrm{of}\;{\dot{\theta }}_{y}$	−0.72	<0.01	-0.65	<0.01	−0.47	<0.01
	$\mathrm{First}\;\mathrm{MR}\;\mathrm{of}\;{\dot{\theta }}_{y}$	−0.61	<0.01	−0.54	<0.01	-	-
	$\mathrm{First}\;\mathrm{RF}\;\mathrm{of}\;\ddot{x}$	−0.56	<0.01	−0.54	<0.01	-	-
	$\mathrm{First}\;\mathrm{MR}\;\mathrm{of}\;\ddot{y}$	-	<0.01	−0.50	<0.01	-	-
Steadiness	$\mathrm{Jerk}\;\mathrm{index}\;\mathrm{of}\;{\dot{\theta }}_{x}$	0.35	0.047	−0.56	<0.01	−0.39	0.027
	Jerk index of MI_AV_	−0.49	<0.01	−0.51	<0.01	-	-
	$\mathrm{Jerk}\;\mathrm{index}\;\mathrm{of}\;\ddot{y}$	−0.49	<0.01	−0.36	0.046	-	-
	Jerk index of MI_LA_	-	-	−0.41	<0.01	-	-
	$\mathrm{Sec}\mathrm{ond}\;\mathrm{RF}\;\mathrm{of}\;{\dot{\theta }}_{x}$	-	-	-	-	−0.41	0.019
	$\mathrm{Sec}\mathrm{ond}\;\mathrm{MR}\;\mathrm{of}\;\ddot{x}$	−0.46	<0.01	-	-	-	-
	$\mathrm{Sec}\mathrm{ond}\;\mathrm{MR}\;\mathrm{of}\;\ddot{z}$	−0.45	<0.01	0.43	0.013	0.44	0.013

**Table 7 sensors-22-07993-t007:** Number of data points for each SCA severity class for the DDKT and FCT in the dataset.

	DDK			FCT		
Trial	Control	Mild SCA	Moderate SCA	Control	Mild SCA	Moderate SCA
Trial 1	9	14	11	9	16	9
Trial 2	10	13	8	9	14	8

**Table 8 sensors-22-07993-t008:** Comparing different classifiers for predicting the severity of DDKT using the features that were extracted from FNT.

Classifier	Metric	Control	Mild SCA	Moderate SCA	Accuracy	Weighted Average
SVC	Recall	0.60	0.84	0.62	0.71	0.71
	Precision	1.0	0.61	0.71		0.76
	Specificity	1.0	0.61	0.91		0.81
	F1-score	0.75	0.71	0.67		0.71
LR	Recall	0.70	0.69	0.62	0.68	0.68
	Precision	0.87	0.64	0.55		0.69
	Specificity	0.95	0.72	0.82		0.82
	F1-score	0.78	0.67	0.59		0.68
GBC	Recall	0.60	0.54	0.75	0.61	0.61
	Precision	0.60	0.64	0.60		0.61
	Specificity	0.81	0.78	0.83		0.80
	F1-score	0.60	0.58	0.67		0.61

**Table 9 sensors-22-07993-t009:** Comparing different classifiers for predicting the severity of FCT using the features that were extracted from FNT.

Classifier	Metric	Control	Mild SCA	Moderate SCA	Accuracy	Weighted Average
SVC	Recall	0.78	0.93	0.62	0.81	0.81
	Precision	0.85	0.76	0.83		0.81
	Specificity	0.95	0.76	0.96		0.87
	F1-score	0.82	0.84	0.71		0.80
LR	Recall	0.22	1.0	0.37	0.61	0.61
	Precision	1.0	0.54	1.0		0.79
	Specificity	1.0	0.30	1.0		0.68
	F1-score	0.36	0.70	0.54		0.56
GBC	Recall	0.55	0.93	0.37	0.68	0.68
	Precision	0.83	0.59	1.0		0.77
	Specificity	0.95	0.47	1.0		0.75
	F1-score	0.67	0.72	0.54		0.66

## Data Availability

The study was an industry-sponsored study and collected data are not available to the public.

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
