# Peer review of "Objective Assessment of Upper-Extremity Motor Functions in Spinocerebellar Ataxia Using Wearable Sensors"

_sensors, 2022, doi:10.3390/s22207993_

Round 1

Reviewer 1 Report

The manuscript (ms) shows practical investigation mainly aiming at objective assessment using commercially available wearable sensors to assess the upper-extremity moto symptoms in spinocerebelllar ataxia (SCA) associated with the standard Assessment and Rating of Ataxia (SARA), for potenial applications for remote diagnostics of SCA. The wearable devices embedded with inertia mass unit (IMU) and gyroscopes showed good performance of the the application of the sensor. Based on the sensing performance, the authors implemented the cyclic test of finger-to-nose-test (FNT), and extract the features and parameters from the sensing device. By comparing the scores of FNT with the other two subjectively tests FCT and DDKT, the authors obtained some correlation between the objective results and the subjective results, thus leading to the conclusion of advantages of the proposed approach. As is written in the ms, larger sampling of test is needed to find practical application.

Overall, I think the ms is well designed in test and the statiscal algorithm is effective to compare the objective and subjective score through 140 features. The figures and tables of data may provide with enough results to support the view of point in the ms. Considering the demonstration and application of the wearable sensor for assessing SCA with objective manner, I think it quite useful and a practical trial, thus it is acceptable for publication in this special issue of Sensors, addressing some minor revision as follows.

1.    How does the sensor work in terms of its working principle? Why can the sensor measure the gesture motion of the patient’s hands in a better and precise manner? I suggest the authors add some solid criterior of the sensing part to facilitate the application in assessing the upper extremity motor.

2.   What is the reason for it to have the advantages in comparison with other conventional assessment approaches? I suggest the authors provide some more discussion in the ms.

3.    What is the relationship between the model and the testament? Please provide more details of the algorithms for varifiying the dependent correlation.

4. As is said by in the ms, the sensor may have been used in the gait and balance in SCA. What are the main difference between the results of testing upper-extremity motion and that of the gait or balance in SCA?

5.    Some of the references are incomplete. Please make sure to modify them.

Author Response

Authors would like to thank both the reviewers to take out time and provide the constructive suggestions to improve the manuscript. We have provided our responses for the reviewers below:

Response to Reviewer 1 Comments

The manuscript (ms) shows practical investigation mainly aiming at objective assessment using commercially available wearable sensors to assess the upper-extremity moto symptoms in spinocerebelllar ataxia (SCA) associated with the standard Assessment and Rating of Ataxia (SARA), for potenial applications for remote diagnostics of SCA. The wearable devices embedded with inertia mass unit (IMU) and gyroscopes showed good performance of the the application of the sensor. Based on the sensing performance, the authors implemented the cyclic test of finger-to-nose-test (FNT), and extract the features and parameters from the sensing device. By comparing the scores of FNT with the other two subjectively tests FCT and DDKT, the authors obtained some correlation between the objective results and the subjective results, thus leading to the conclusion of advantages of the proposed approach. As is written in the ms, larger sampling of test is needed to find practical application.

Overall, I think the ms is well designed in test and the statiscal algorithm is effective to compare the objective and subjective score through 140 features. The figures and tables of data may provide with enough results to support the view of point in the ms. Considering the demonstration and application of the wearable sensor for assessing SCA with objective manner, I think it quite useful and a practical trial, thus it is acceptable for publication in this special issue of Sensors, addressing some minor revision as follows.

Point 1: How does the sensor work in terms of its working principle? Why can the sensor measure the gesture motion of the patient’s hands in a better and precise manner? I suggest the authors add some solid criterior of the sensing part to facilitate the application in assessing the upper extremity motor.

Response 1: We appreciate constructive suggestion from the reviewer. Following text is added in the manuscript (Line 65-69):

The working principle of these IMUs is based on estimating movement using onboard sensors, including an accelerometer, magnetometer, or gyroscope. The benefit of IMUs over other sensing modalities is in precise measurement of angular acceleration and velocity with minimal preparation and no interference with the motor examination.”

Point 2: What is the reason for it to have the advantages in comparison with other conventional assessment approaches? I suggest the authors provide some more discussion in the ms.

Response 2: We have added following text in the discussion section (Line 339-343):

“The benefit of the proposed objective assessment over conventional subjective evaluation includes precise measurement of movement. Due to high sensitivity in tracking hand movement and detecting smaller vibrations (i.e., tremors), the proposed solution may identify subtle motor symptoms, which are often challenging to detect visually.”

Point 3: What is the relationship between the model and the testament? Please provide more details of the algorithms for varifiying the dependent correlation.

Response 3: The model was used to predict the SCA severity. For this purpose, we used the the Scale for the Assessment and Rating of Ataxia (SARA) score as the dependent variable for the model. The model used sensor-derived parameters related to the timing, variability, speed, rhythmicity, and steadiness as the input features.

Following statement has been added in the method section give more details about the analyses (Line 297 to 300):

  “Although each motor task (i.e., FNT, FCT, or DDKT) provides unique information about movement dysfunction, there might be an inherent redundancy among these motor tasks [19, 20]. The purpose of estimating the correlations was to capture the in-herent redundancies among the tests (i.e., FNT. FCT and DDKT) through correlation analyses.”

Point 4: As is said by in the ms, the sensor may have been used in the gait and balance in SCA. What are the main difference between the results of testing upper-extremity motion and that of the gait or balance in SCA?

Response 4: We have included following text in the discussion section (Line number: 344 to 350):

Unlike previous studies which used wearable-based validated gait assessment tools to assess SCA severity [11, 19], the present study focused on upper-extremity motor examination by decomposing arm movement into discrete subcomponents using the wearable. This is a novel and valuable approach for representing and quantifying the ataxia phenotype as gait assessment can be only administered for ambulatory individuals and require dedicated facilities (e.g., uninterrupted walk pathway, or trained staff to ensure safety). The present study would contribute to validating the wearable-based upper extremity motor assessment to assess SCA severity.”

Point 5: Some of the references are incomplete. Please make sure to modify them.

Response 5: We have carefully reviewed the accuracy of the cited references and corrected them according to the guidelines of the journal.

nson's disease).

Reviewer 2 Report

This paper reports a method to track spinocerebellar ataxia (SCA) symptoms using a wearable accelerometer sensor, which collect datas from the patients participants. By using the algorithm to extract the features of finger-to-nose test, SCA symptoms can be predicted. The overall quilty of this paper is considerable high. However, I do have some questions and comments about this paper. 

Major comments:

(1) About the innovation of this paper. There have been similar works using wearable sensors to assess Spinocerebellar Ataxia (Movement Disorders, 2021)[R1], I think there are no obvious differences in the methods used in the two articles. The authors should make a comparison with similar research and explain the biggest advantages and innovations of this work.

(2) Some questions about sensors. There is too little information about the wearable sensors, such as the working principle of the sensor, which data can be measured by the sensor. In addition, the authors did not test whether small displacement in sensor installation positions could cause differences in measurement results, which is important for the accuracy of the assessment.

(3) The authors mentioned building predictive models to estimate the severity of symptoms, but the paper did not explain how to build a model to automatically estimate the severity of SCA based on the data measured by sensors..

Minor comments:

(1) There are some problems with the table: there are two Steadiness in Table 1, FC in Table 6 and Table 7 should be FCT.

(2) In addition to Spinocerebellar ataxia, there are other diseases that can cause similar upper extremity movement disorders, such as cervical spondylotic myelopathy and Parkinsonism, and these should be distinguished.

References:

[1] Shah, V.V. et al. Gait Variability in Spinocerebellar Ataxia Assessed Using Wearable Inertial Sensors. Movement Disorders, 2021, 36 (12) : 2922-2931.

Author Response

Authors would like to thank both the reviewers to take out time and provide the constructive suggestions to improve the manuscript. We have provided our responses for the reviewers below:

Response to Reviewer 2 Comments

This paper reports a method to track spinocerebellar ataxia (SCA) symptoms using a wearable accelerometer sensor, which collect datas from the patients’ participants. By using the algorithm to extract the features of finger-to-nose test, SCA symptoms can be predicted. The overall quality of this paper is considerable high. However, I do have some questions and comments about this paper.

Point 1: About the innovation of this paper. There have been similar works using wearable sensors to assess Spinocerebellar Ataxia (Movement Disorders, 2021) [R1], I think there are no obvious differences in the methods used in the two articles. The authors should make a comparison with similar research and explain the biggest advantages and innovations of this work.

References:

[1] Shah, V.V. et al. Gait Variability in Spinocerebellar Ataxia Assessed Using Wearable Inertial Sensors. Movement Disorders, 2021, 36 (12) : 2922-2931.

Response 1: We have included following text in the discussion section (Line number: 344 to 351):

Unlike previous studies which used wearable-based validated gait assessment tools to assess SCA severity [11, 19], the present study focused on upper-extremity motor examination by decomposing arm movement into discrete subcomponents using the wearable. This is a novel and valuable approach for representing and quantifying the ataxia phenotype as gait assessment can be only administered for ambulatory individuals and require dedicated facilities (e.g., uninterrupted walk pathway, or trained staff to ensure safety). The present study would contribute to validating the wearable-based upper extremity motor assessment to assess SCA severity.”

Point 2: Some questions about sensors. There is too little information about the wearable sensors, such as the working principle of the sensor, which data can be measured by the sensor. In addition, the authors did not test whether small displacement in sensor installation positions could cause differences in measurement results, which is important for the accuracy of the assessment.

Response 2: We appreciate constructive suggestion from the reviewer. Following text is added in the manuscript (Line 65-69):

The working principle of these IMUs is based on estimating movement using onboard sensors, including an accelerometer, magnetometer, or gyroscope. The benefit of IMUs over other sensing modalities is in precise measurement of angular acceleration and velocity with minimal preparation and no interference with the motor examination.”

We understand and appreciate the concern raised by the reviewer. The tests were performed in the supervised condition, based on study guidelines, which specified the sensor's location. The research staff was trained on the study protocol to ensure that the sensor was located in a current position before data collection. This is consistent with other instrumented assessments using wearable sensors such as gait and balance analysis. We acknowledge that the future use of the system in the home and unsupervised setting would require additional validation and algorithm developments to account for sensor placement. To clarify it further, we added the following in the method section (Line 120-121):

The sensors were secured over the wrist using an elastic Velcro to avoid any shift in the sensor placement.”

Point 3: The authors mentioned building predictive models to estimate the severity of symptoms, but the paper did not explain how to build a model to automatically estimate the severity of SCA based on the data measured by sensors.

Response 3: We appreciate this comment and have removed the corresponding statements from the manuscript. By performing a much larger study, it would be possible to extend the study to develop an automated predictive system of disease severity. However, this is outside the scope of the current manuscript.

Point 4: There are some problems with the table: there are two Steadiness in Table 1, FC in Table 6 and Table 7 should be FCT.

Response 4: We have corrected the text as suggested by the Reviewer.

Point 5: In addition to Spinocerebellar ataxia, there are other diseases that can cause similar upper extremity movement disorders, such as cervical spondylotic myelopathy and Parkinsonism, and these should be distinguished.

Response 5: We agree with the reviewer. This would be a subject of future studies to compare the platform's accuracy in differentiating ataxia from other neurodegenerative diseases (e.g., cervical spondylitis myelopathy and Parkinson's disease).
